# Image Translation by Ad CycleGAN for COVID-19 X-Ray Images: A New Approach for Controllable GAN

**DOI:** 10.3390/s22249628

**Published:** 2022-12-08

**Authors:** Zhaohui Liang, Jimmy Xiangji Huang, Sameer Antani

**Affiliations:** 1Information Retrieval and Knowledge Management Laboratory, York University, Toronto, ON M3J 1P3, Canada; 2National Library of Medicine, National Institutes of Health, Bethesda, MD 20894, USA

**Keywords:** generative adversarial networks, applied machine learning, X-ray images, digital health in the midst of COVID-19

## Abstract

We propose a new generative model named adaptive cycle-consistent generative adversarial network, or Ad CycleGAN to perform image translation between normal and COVID-19 positive chest X-ray images. An independent pre-trained criterion is added to the conventional Cycle GAN architecture to exert adaptive control on image translation. The performance of Ad CycleGAN is compared with the Cycle GAN without the external criterion. The quality of the synthetic images is evaluated by quantitative metrics including Mean Squared Error (MSE), Root Mean Squared Error (RMSE), Peak Signal-to-Noise Ratio (PSNR), Universal Image Quality Index (UIQI), visual information fidelity (VIF), Frechet Inception Distance (FID), and translation accuracy. The experimental results indicate that the synthetic images generated either by the Cycle GAN or by the Ad CycleGAN have lower MSE and RMSE, and higher scores in PSNR, UIQI, and VIF in homogenous image translation (i.e., *Y* → *Y*) compared to the heterogenous image translation process (i.e., *X* → *Y*). The synthetic images by Ad CycleGAN through the heterogeneous image translation have significantly higher FID score compared to Cycle GAN (*p* < 0.01). The image translation accuracy of Ad CycleGAN is higher than that of Cycle GAN when normal images are converted to COVID-19 positive images (*p* < 0.01). Therefore, we conclude that the Ad CycleGAN with the independent criterion can improve the accuracy of GAN image translation. The new architecture has more control on image synthesis and can help address the common class imbalance issue in machine learning methods and artificial intelligence applications with medical images.

## 1. Introduction

Coronavirus disease 2019 (COVID-19) is an infectious disease caused by a novel coronavirus SARS-CoV-2. The most common clinical manifestation of COVID-19 infection is a specific type of pneumonia which rapidly leads to severe acute respiratory infection symptoms and may even directly develop into acute respiratory distress syndrome (ARDS) [1]. Diagnostic methods for COVID-19 include new medical technologies from various domains. Although the gold standard for confirmation is the real-time reverse-transcriptase polymerase chain reaction (RT-PCR), the test sensitivity is about 96.0%, and its performance is affected by the disease prevalence in the given population [2]. The diagnosis, therefore, is a combination of RT-PCR test result with various clinically accessible methods such as contact history, physical examination, and radiographic imaging. The radiological diagnostic methods include imaging using computed tomography (CT), chest X-ray (CXR), and lung ultrasound (LUS), etc. While not a primary step, they still play important roles in confirming and staging positive cases. During the pandemic, many research groups have collected relevant medical images to develop new artificial intelligence (AI) technologies for automated COVID-19 screening and diagnosis [3], particularly applying deep neural networks (DNNs) for detecting image patterns consistent with the disease [4]. However, the effectiveness and generalizability of the methods are adversely impacted by the lack of sufficiently large number of adequately labeled COVID-19 images examples to build a balanced training set. A DNN trained using an imbalanced dataset, where cases only occupy 5% to 6% of the total image samples, will reach a performance threshold earlier than its theoretical capacity determined by the architecture [5]. A study published in 2020 revealed that the seemly high-performing DNN models for COVID-19 detection in CXR images are vulnerable to network attacks [6]. Another challenge is that the specific medical image patterns are different from general-purposed images such as those in the ImageNet dataset. When using transfer learning with DNN models trained with ImageNet to fine tune a new model for the radiography images, the pretrained feature extractors usually cannot effectively capture the medical significant patterns through the complex architecture but simply develop meaningless combinations for the final decision. All these factors contribute to the vulnerability of the current DNN technology for COVID-19 image pattern recognition and detection.

Image translation is a common task of image synthesis supported by generative adversarial networks (GAN) [7] and dual learning [8]. The objective of image translation is to learn the mapping between two image domains by dual learning with GAN. The current benchmark methods for image translation are the Pix2Pix for paired images learning [9] and cycle-consistent generative adversarial network (Cycle GAN) for unpaired images learning [10]. Image translation is widely used in medical image applications for cross-modality image synthesis with multiple purposes, such as image registration, data augmentation for improving model generalization capacity for image detection, classification, and segmentation, etc. Popular methods are based on the Cycle GAN architecture for image translation between medical images by different technologies [11,12,13]. The appearance of a radiologic image is like a gray-scale image, but different imaging mechanisms in fact result in the anatomical and pathological patterns being exhibited differently across computed tomography (CT), magnetic resonance (MR), and positron emission tomography (PET) images. With a deep neural network (DNN) architecture, GAN can learn the detailed mapping between two medical imaging pattern domains. With an optimized GAN, the digital images acquired from different methods (i.e., CT, MR, or PET) can be effectively converted to each other. This function helps the radiologists to maximize their performance to interpret the clinical findings without asking the patients to do all types of examinations. Furthermore, the new technology helps to lower the radiational dose for the examination to protect the patients while keeping the best diagnostic performance.

The most common application of medical images translation is to convert between CT and MR images. For example, Fu J. et al. introduced the sCTcycleGAN to convert MR images to CT images [14]. Lee J.H. et al. [15] and Hu N. et al. [16] used cGAN models to perform conversion of CT image to MR images to acquire more detailed information. Conversely, Nie D et al. [17] and Emami H et al. [18] used serialized GAN models to implement MR images to CT images conversions. Another type of image translation is to convert PET images to CT images, such as the study by Hu Z et al. using a WGAN model to perform attenuation correction and to convert PET to pseudo-CT images [19]. Bazangani F. et al. introduced an E-GAN for translating 3D FDG-PET image to MR image [20]. The main purpose of these image translation applications is to convert the radiologic images from a complex format to relatively simple format, such as from PET to MR, then from MR to CT, because the latter ones are easier to interpret by empirical medical expertise. However, one interesting topic of image translation is seldom involved, i.e., to perform the image domain translation from the normal domain to a certain disease domain. This idea is intuitive because a medical image with some morbid abnormality can be interpreted as: normal patterns + disease patterns. This type of application will make great contributions for rare or newly discovered diseases when the images containing the disease patterns are difficult to acquire while the images with the corresponding normal structure are accessible. A typical example is the COVID-19 radiological images acquired during the beginning of the pandemic. Thus, it became the principle of our experimental design.

The main contribution of this study is to introduce an external criterion to the current state-of-the-art GAN architecture (Cycle GAN) for image translation, which can ensure the generated synthetic images belonging to both the correct image domain and the correct diagnosis class. This design will be easy to extend to other medical or non-medical data synthetization applications. The rest of this paper is organized as follows. In Section 2 Materials and Methods, we present the rationale for cycle-consistent adversarial network (Cycle GAN) and its restrictions which can be solved by the new adaptive Cycle GAN (Ad CycleGAN) architecture. The pseudo code for the Ad CycleGAN optimization and the quantitative evaluation metrics are also presented in this section. In Section 3 Experiments, we present the Ad CycleGAN experiment based on the open-source COVID-19 image dataset and its performance compared with the conventional Cycle GAN. In Section 4 Discussion and Section 5 Conclusions, we combine the findings of the experiments in Section 3 with the Ad CycleGAN design discussed in Section 2 to summarize the merits of new Ad CycleGAN and explore its potential applications in biomedicine.

## 2. Materials and Methods

### 2.1. Cycle-Consistent Adversarial Network and Its Restriction

Cycle-consistent adversarial network, or Cycle GAN, is the state-of-the-art conditional generative adversarial network (CGAN) for unpaired image to image translation. A typical Cycle GAN uses two generators and two discriminators to learn the mapping of two distributions by optimizing with a complex objective and reaching a state of adversarial equilibrium. During optimization, the objective of the Cycle GAN has three components: adversarial loss, cycle consistency loss, and identity loss. The adversarial loss follows the original GAN design to measure the difference of the generated images and the target images. To mapping between two image distribution *X* and *Y* with Cycle GAN, we use two pairs of generators and discriminators in the Cycle GAN model. The first pair G and $D_{Y}$, aims to adversarially generate and distinguish the images belong to domain *X* and the images belonging to domain *Y*, i.e., $min_{G}max_{D_{Y}}L_{GAN}\left(G,D_{Y},\;X,Y\right)$. The optimization objective is written as:(1)$$L_{GAN}\left(G,D_{Y},\;X,Y\right)=E_{y\sim p_{data}\left(y\right)}\left[\log D_{Y}\left(y\right)\right]+E_{x\sim p_{data}\left(x\right)}\left[\log\left(1-D_{Y}\left(G\left(x\right)\right)\right)\right]$$
where the data distribution of *X* and *Y* are denoted as $x\sim p_{data}\left(x\right)$, and $y\sim p_{data}\left(y\right)$. On the other hand, the second pair F and $D_{X}$, aims to adversarially generate and distinguish the generated and real images belonging to domain *Y*, i.e., $min_{F}max_{D_{X}}L_{GAN}\left(F,D_{X},\;Y,X\right)$, the corresponding objective is writing as:(2)$$L_{GAN}\left(F,D_{X},\;Y,X\right)=E_{x\sim p_{data}\left(x\right)}\left[\log D_{X}\left(x\right)\right]+E_{y\sim p_{data}\left(y\right)}\left[\log\left(1-D_{X}\left(F\left(y\right)\right)\right)\right]$$

The total adversarial loss during a single iteration is the summation of the loss from Equations (1) and (2), i.e., $L_{GAN}\left(G,D_{Y},\;X,Y\right)+L_{GAN}\left(F,D_{X},\;Y,X\right).$ The adversarial optimization can theoretically learn the mappings of *G* and *F* to produce identical images. However, this ideal outcome is unrealistic in two ways. First, given the ideal situation, the generator networks can randomly map an input image from the source domain to a random image in the target domain, which is not our desired outcome. Second, we need to guarantee that the generated images have valid shapes and other uncommon elements within a reasonable scope of the real images. We add the cycle-consistent losses to reversely translate the images back to their original domains, i.e., $x\to G\left(x\right)\to F\left(G\left(x\right)\right)\approx x$ (forward cycle consistency), and $y\to F\left(y\right)\to G\left(F\left(y\right)\right)\approx y$ (backward cycle consistency). The total cycle-consistent loss is written as:(3)$$L_{cyc}\left(G,F\right)=E_{x\sim p_{data}\left(x\right)}\left[F\left(G\left(x\right)\right)-x_{1}\right]+E_{y\sim p_{data}\left(y\right)}\left[G\left(F\left(y\right)\right)-y_{1}\right]$$

Furthermore, an identity loss is added to measure how close the generated image to the real image itself if the real image goes through the Cycle GAN generator, i.e., $x\to F\left(x\right)\approx x$, and $y\to G\left(y\right)\approx y$. Adding to identity loss to the total loss of the generator can help to preserve the original color. The identity loss can be expressed as:(4)$$L_{iden}\left(G,F\right)=E_{x\sim p_{data}\left(x\right)}\left[F\left(x\right)-x_{1}\right]+E_{y\sim p_{data}\left(y\right)}\left[G\left(y\right)-y_{1}\right]$$

The full generator loss of the Cycle GAN is written as the summation of the above three loss functions:(5)$$L_{total}=L_{GAN}\left(G,D_{Y},\;X,Y\right)+L_{GAN}\left(F,D_{X},\;Y,X\right)+\lambda L_{cyc}\left(G,F\right)+\sigma L_{iden}\left(G,F\right)$$
where λ and σ are the parameters to respectively adjust the importance of the cycle-consistent loss, and the identity loss during the model optimization. Thus, the objective for the whole model optimization is to solve:(6)$$G^{*},\;F^{*}=\arg min_{G,F}\;max_{D_{X},D_{Y}}L\left(G,F,D_{X},D_{Y}\right)$$

In our experiments, we find that though the Cycle GAN produces visually plausible synthetic images, it cannot guarantee the synthetic images to be classified to the correct category by an independently optimized DNN model. In general, the synthesized medical images by Cycle GAN have three drawbacks. First, the generator cannot produce images with high complexity due to the imbalanced information of the two image domains. Second, the translation from a domain with rich information to another domain with relatively poor information (e.g., translating MR images to CT images) is likely to cause ambiguity mapping, which means there can be multiple alternatives in the target domain corresponding to the identical input in the source domain. Third, the Cycle GAN is easily diverted by some improper constrains due to its unpaired image translation setting. The random encoded latent information not only distracts the model from ideal image translation, but even makes the model sensitive to disturbances and variations of input samples [12].

To solve the above problems, we propose the Adaptive Cycle GAN model, or Ad CycleGAN with external criterion to reduce the negative influence of random noise during the Cycle GAN optimization. We believe this new design can effectively improve both the quality of the synthetic images and the accuracy of the synthetic images to the target domain through the image translation process.

### 2.2. External Criterion in Cycle GAN Optimization

The term criterion originates from the concept of critics introduced by Arjovsky M et al. for the optimization of their Wasserstein GAN (WGAN) model [21]. Unlike the original GAN by Goodfellow I. et al. [7] using the discriminator to estimate the likelihood whether the synthetic images by the generator to be true, the WGAN uses the discriminator as a critic to evaluate the quality of the generated images against the real ones. We extend this idea by adding a pre-trained DNN as an independent criterion to evaluate the generated images from multiple aspects except simply classifying as real or fake. The errors from different criteria can be finally congregated as the criterion loss as a new component of the total generator loss. In the Ad CycleGAN model, we introduce two loss terms: the cycle criterion loss and the identity criterion loss. Both are estimated by a pretrained residual network for the likelihood of the synthetic images to the correct category. The joint criterion loss is written as:(7)$$L_{c}=L_{c-cycle}+L_{c-identity}$$
where the cycle criterion loss is to measure the similarity of $x\;\sim \;F\left(G\left(x\right)\right)$ and of $y\;\sim \;G\left(F\left(y\right)\right)$, and the identity criterion loss is to measure the similarity of $x\;\sim \;F\left(x\right)$ and $y\;\sim \;G\left(y\right)$. In other words, like the cycle loss and identity loss, the cycle criterion loss $L_{c-cycle}$ quantitatively measures whether of the back-translated images are still be classified as the original class. The identity criterion loss $L_{c-identity}$ quantitatively measures whether the trained generators can produce real images from a real observed sample that still consistent to the same class.

In addition, when the GAN training reaches an adversarial equilibrium, the criterion loss can periodically add an extra oscillation momentum to the stable condition to push the generator progress to learn more details. The new $L_{c}$ term is considered as a regularization method to prevent the saturated status of the GAN optimization because it provides a method to make the GAN training controllable to a certain degree. However, we need to add an empirical decay factor to the criterion loss term to control its side effect of breaking the adversarial equilibrium leading the GAN model to learn the loss patterns again through more iterations.

### 2.3. Ad CycleGAN Architecture

The Ad CycleGAN consists of two pairs of generators and discriminators to learning the mapping between the image domain, and a pre-trained independent criterion to ensure the generated images containing the key discriminative patterns for the two image domains. The total loss function of the generators in the Ad CycleGAN consists of four parts:

Adversarial loss: $L_{GAN}\left(G,D_{Y},\;X,Y\right)+L_{GAN}\left(F,D_{X},\;Y,X\right)$Cycle consistency loss: $L_{cycle}\left(G,F\right)$Identity loss: $L_{iden}\left(G,F\right)$Criterion Loss: $L_{c}=L_{c-cycle}+L_{c-identity}$Thus, the total generator loss in Equation (5) is revised as:(8)$$L_{total}=[L_{GAN}\left(G,D_{Y},\;X,Y\right)+L_{GAN}\left(F,D_{X},\;Y,X\right)]+\lambda L_{cycle}\left(G,F\right)+\lambda L_{iden}\left(G,F\right)+\kappa (\varphi L_{c})$$

In the experiment for the COVID-19 X-ray image synthesis, the generators follow the U-Net architecture [22] with skip connections to reduce the input feature size from 64 by 64 to 1 by 1 then restore to 64 by 64. The discriminators follow the PatchGAN architecture [9] with an output of 4-by-4-by-1 feature map (given the low resolution of our dataset) to determine with the images are real or fake. We choose to use the binary cross entropy as the objective function for the discriminator loss and the adversarial loss terms for the generators. The cycle consistency loss and the identity loss use the mean of absolute error (MAE) function as the objective. The external criterion is a residual network with three residual modules. Each residual module has three convolutional layers with a skip connection from the first convolutional layers to the third one to ensure gradient flow when optimized by backpropagation. The residual modules are connected by batch normalization and max pooling layers to accordingly reduce the tensor size. It uses the sparse categorical cross entropy function as the loss objective, and it is optimized by the adaptive moment estimation (Adam) algorithm with the initial learning rate of $1\times 10^{-4}$ with 100 epochs. During the GAN optimization, the pretrained criterion estimates the input images with the output logits which can be combined to other loss terms. The terms of the criterion loss are measured by the sparse categorical cross entropy as the same method as how the pretrained criterion was optimized. Though some studies recommend using the unbounded smooth loss function such as to optimize the GAN models such as Wasserstein Loss or Mean Square Error (MSE) [21,23]. Empirically, the choice of loss functions is mainly based on the components of the total loss objective. If all errors can be measured within similar scales, using the unbounded loss functions is straightforward and easier for the overall GAN optimization. However, if the GAN architecture consists of many components like this case, using hypermeters to adjust the importance of different terms or to determine the frequency of loss injection to the total loss can provide a more flexible option for GAN optimization as described in Equation (8). The Ad Cycle GAN architecture is illustrated in Figure 1. Additionally, the pseudo code of the optimization algorithm for the Ad CycleGAN model is presented in Algorithm 1.
**Algorithm 1.** Ad CycleGAN Optimization1:**for** number of epochs **do**2:     **for** number of batches **do**3:         Sample minibatch $\leftarrow {\left\{x^{\left(i\right)}\right\}}_{i=1}^{m}\in X$
4:Sample minibatch $\leftarrow {\left\{y^{\left(j\right)}\right\}}_{j=1}^{m}\in Y$
5:         Generate m synthetic samples of $G\left(x\right)and\;F\left(y\right)$
6:$\;\;\;\;\;\;\;\;\;\;\;\;\;\;\;\;\;\;\;synthetic\;X:\;X\;\to G\left(x\right)$7:$\;\;\;\;\;\;\;\;\;\;\;\;\;\;\;\;\;\;\;synthetic\;Y:\;Y\;\to F\left(y\right)$8:         Compute the Adversarial loss9:$\;\;\;\;\;\;\;\;L_{GAN}\left(G,D_{Y},\;X,Y\right)=E_{y\sim p\left(y^{\left(m\right)}\right)}\left[\log D_{Y}\left(y\right)\right]+E_{x\sim p\left(x^{\left(m\right)}\right)}\left[\log\left(1-D_{Y}\left(G\left(x\right)\right)\right)\right]$10:$\;\;\;\;\;\;\;\;L_{GAN}\left(F,D_{X},\;Y,X\right)=E_{x\sim p_{data}\left(x\right)}\left[\log D_{X}\left(x\right)\right]+E_{y\sim p_{data}\left(y\right)}\left[\log\left(1-D_{X}\left(F\left(y\right)\right)\right)\right]$11:         Generate m cycle sample of $F\left(G\left(x\right)\right)and\;G\left(F\left(y\right)\right)$
12:$\;\;\;\;\;\;\;\;\;\;\;\;\;\;\;\;\;\;\;Cycle\;X:\;G\left(x\right)\to F\left(G\left(x\right)\right)$13:$\;\;\;\;\;\;\;\;\;\;\;\;\;\;\;\;\;\;\;Cycle\;Y:\;F\left(y\right)\to G\left(F\left(y\right)\right)$14:         Compute the Cycle loss15:$\;\;\;\;\;\;L_{cyc}\left(G,F\right)=E_{x\sim p\left(x^{\left(m\right)}\right)}\left[F\left(G\left(x\right)\right)-x_{1}\right]+E_{y\sim p\left(y^{\left(m\right)}\right)}\left[G\left(F\left(y\right)\right)-y_{1}\right]$16:         
Generate m identical sample of $F\left(x\right)and\;G\left(y\right)$
17:$\;\;\;\;\;\;\;\;\;\;\;\;\;\;\;\;\;\;\;identical\;X:X\;\to F\left(x\right)$18:$\;\;\;\;\;\;\;\;\;\;\;\;\;\;\;\;\;\;\;identical\;Y:Y\;\to G\left(y\right)$19:         Compute the identity loss20:$\;\;\;\;\;\;L_{iden}\left(G,F\right)=E_{x\sim p\left(x^{\left(m\right)}\right)}\left[F\left(x\right)-x_{1}\right]+E_{y\sim p\left(y^{\left(m\right)}\right)}\left[G\left(y\right)-y_{1}\right]$21:         Compute the criterion loss for cycle sample: $L_{c-cycle}$
22:         Compute the criterion loss for identical sample: $L_{c-identity}$
23:         Compute the total generator loss24:$\;\;\;\;\;L_{total}=[L_{GAN}\left(G,D_{Y},\;X,Y\right)+L_{GAN}\left(F,D_{X},\;Y,X\right)]+\lambda L_{cycle}\left(G,F\right)+\lambda L_{iden}\left(G,F\right)+\kappa (\varphi L_{c})$25         Update the Discriminator $D_{X}\;and\;D_{Y}$
26:$\;\;\;\;\;\;\;\;\;\;\;\;\;\;\;\;\;\;\;max_{D_{X}}L_{GAN}\left(F,\;D_{X},\;X,\;Y\right)$27:$\;\;\;\;\;\;\;\;\;\;\;\;\;\;\;\;\;\;\;max_{D_{Y}}L_{GAN}\left(G,\;D_{y},X,\;Y\right)$28:          Update the Generators G, F
29:$\;\;\;\;\;\;\;\;\;\;\;\;\;\;\;\;\;\;\;min_{G,F}L\left(G,F,D_{X},\;D_{Y}\right)$30:   **end do**
31:
 **end do**


### 2.4. Evaluation Metrics for Translated Images

The performance evaluation of GAN networks is usually subjective and remains as an open problem [24]. Our objective is to generate synthetic medical images with good fidelity and diversity. We need to measure both the quality of the images and ensure the generated images belonging to the correct category, i.e., carrying the diagnostically significant patterns. The latter task can be measured by the classification accuracy of the synthetic images. There are generally two types of methods to measure the quality of the synthetic images: subjective evaluation and objective evaluation. Subjective evaluation requires human expertise. It is time consuming and difficult to replication. Therefore, we apply the objective metrics to compare the synthetic images and the generated images with the assumption that the high-quality synthetic images have higher degrees of the similarity to the real images. The quantitative evaluation metrics for our experiments include Mean Squared Error (MSE), Root Mean Squared Error (RMSE), Peak Signal-to-Noise Ratio (PSNR), Universal Image Quality Index (UIQI), and Visual Information Fidelity (VIF).MSE, RMSE and PSNR are metrics to measure the pixel difference between the synthetic images and the real images. MSE is the accumulated mean squared error of two images, and RMSE is the accumulated root mean square error of the two images. PSNR is a measure for image quality [25] based on the pixel difference between the synthetic image and the real image. UIQI summarizes the attributes of human vision [26], where synthetic images and the real images are compared in three aspects: luminance, contrast, and structure. VIF is another measure based on human visual perception. VIF quantifies the image fidelity by the difference of the information extracted from the real image and the information loss to the synthetic image by human brain is quantified as the VIF score using visual natural scene statistics (NSS), human visual system (HVS) and an image distortion model. For comparison, the synthetic images with low MSE and RMSE, and with high scores in PSNR, UIQI and VIF are considered to be of better quality. In addition, we also use the Frechet Inception Distance (FID) which is a commonly accepted metric to compare the quality of the images synthesized by different generative models. FID was proposed by Heusel, M. et al. in 2017 to calculate the distance between feature vectors calculated for real and generated images [27]. It reflects how similar the two image groups are in terms of statistics on computer vision features of the raw images calculated using a pretrained classifier. Low FID score indicates the two groups of images are similar or have more similar statistics.In the next section, we will present the experiments of respectively using Cycle GAN and Ad CycleGAN to perform image translation between normal CXR images and COVID-19 positive CXR images, and the comparisons of the quality of the synthetic with the above quantitative metrics.

## 3. Experiments

### 3.1. Material and Methods

In our experiments, we respectively implemented the Cycle GAN and the Ad CycleGAN to perform image translation between normal and COVID-19 positive CXR images. According to the clinical observation, the COVID-19 positive cases have special bilateral or unilateral multiple mottling and ground-glass opacity patterns on the CXR and CT images [1] and these patterns have been successfully captured by multiple DNN models [3,4,5]. Based on the above discussion, the image translation between normal and COVID-19 positive images can be formulated as learning the mapping to add or to remove such diagnostic significant patterns between the normal and the COVID-19 image domains. We use an image dataset consisting of 219 COVID-19 positive images, and 1064 normal CXR from the Kaggle COVID-19 Radiography dataset (https://www.kaggle.com/datasets/tawsifurrahman/covid19-radiography-database, accessed on 29 August 2022) for the experiments. Figure 2 illustrates some example images in the dataset.

Given the hardware condition, the images are resized to 64-by-64, 3 channel as the input dimensions. Because we have only 219 real COVID-19 X-ray images, 50 of the COVID-19 images are randomly selected and withheld for testing, the rest 169 real images are duplicated 6 times to match 1014 normal X-ray images for model optimization. The Cycle GAN and Ad CycleGAN models are respectively optimized by 600 epochs on the Google Colab platform with GPU. The average runtime is about 58 s per epoch with the mini-batch size of 64.

### 3.2. Results and Interpretation

As shown in Equation (8), we respectively optimized the Cycle GAN and the Ad CycleGAN with similar parameter configurations, with λ=80.0, σ=60.0, φ=0.1, and κ=20. It means that in the optimization of Ad CycleGAN, the criterion loss is added to the total generator loss term very 20 steps. The models are optimized by the Adam optimizer with the initial learning rate of $2\times 10^{-4}$ for 600 epochs. The mini-batch size is 64. The synthetic CXR images respectively generated by the Cycle GAN and by the Ad CycleGAN are shown in Figure 3 and Figure 4. Note the both Cycle GAN and Ad Cycle GAN can perform heterogenous translation, which means the normal images are translated to COVID-19 positive images and vice versa (i.e., X→Y or Y→X); and they can also perform homogeneous translation, which converts the input images within the same domain (i.e., X→X or Y→Y). The homogeneous translation is considered as image augmentation by GAN.

The quantitative measures for the quality of the synthetic images are listed in Table 1, the FID score, and the classification accuracy to the due category of the synthetic images are listed in Table 2.

From Figure 3 and Figure 4, we observe that both Cycle GAN and Ad CycleGAN can synthesize high quality COVID-19 X-ray images with good visual fidelity and diversity through the image translation. Another finding is that both Cycle GAN and Ad CycleGAN can not only perform image translation, but also convert the input images aligned on the sagittal axis to the synthetic images aligned on the coronal axis. The quantitative metrics indicate that the synthetic images generated either by the Cycle GAN or by the Ad CycleGAN have lower MSE and RMSE, and higher scores in PSNR, UIQI, and VIF through the image augmentation process (i.e., Y→Y) compared to the image translation process (i.e., X→Y). It implies the GANs cannot translate high quality synthetic images probably due to insufficient training samples.

The synthetic images by Ad CycleGAN through the heterogeneous image translation (i.e., X→Y) have significantly higher FID score compared to Cycle GAN (p<0.01). However, Cycle GAN generates comparable or even slightly better images through the homogeneous translation or augmentation process (i.e., Y→Y). The image translation accuracy of Ad CycleGAN is higher than the Cycle GAN when the normal images are converted to COVID-19 positive images (p<0.01). However, both Ad CycleGAN and Cycle GAN can perfectly perform homogeneous translation with the accuracy of 100%., i.e., augmenting the image diversity within the COVID-19 positive image domain. It implies that the independent criterion in the Ad CycleGAN can improve the accuracy for heterogeneous image translation.

In our literature review, the similar research on COVID-19 CXR image synthesis mainly uses accuracy as the main metric for GAN performance. For example, Motamed S et al. used GAN as data augmentation to improve the COVID-19 CXR image classification accuracy from 0.81 to 0.84 [28]. Morís DI et al. used CycleGAN to improve COVID-19 CXR image screening accuracy to about 0.90 [29]. Both studies did not use GAN with effective controllable mechanism to guarantee the synthetic images falling into the COVID-19 positive category.

When observing the loss change through the Ad CycleGAN optimization process in Figure 5, we find the influence of the independent periodic criterion loss (added to the total loss every 20 steps) to the total GAN optimization. The criterion loss impacts on both image translation directions at the beginning of the GAN optimization in the early 200 epochs, then it becomes stable afterwards. However, the total loss for discriminator *Y* drops approximately at the last 50 epochs. It implies the external criterion exerts more impact on synthesized COVID-19 positive images, which explains why the image translation by Ad CycleGAN has higher accuracy than that by Cycle GAN.

## 4. Discussion

We present the study of Ad CycleGAN for image translation between normal chest X-ray images and COVID-19 positive chest X-ray images. The experiment compared the performance of new Ad CycleGAN with the conventional Cycle GAN with a series of quantitative metrics of image quality, the FID score, and image translation accuracy. The results indicate that Ad CycleGAN generates synthetic COVID-19 images with higher accuracy than those generated by Cycle GAN. When performing heterogenous image translation, the Ad CycleGAN can generate synthetic COVID-19 positive images with higher FID score and accuracy. Therefore, we conclude the adaptive external criterion design of the Ad CycleGAN can effectively control the image category for image translation. The Ad CycleGAN is considered as a new approach of conditional GAN which can extend the control power upon the synthetic image domain for GAN image synthesis and translation.

The proposed Ad CycleGAN in this study follows the GAN optimization strategy originated from the Wasserstein GAN (WGAN) [21], where the objective is not to estimate the probability of the synthetic images considered as real images, but to “rate” the synthetic images as an objective critic. Under this framework, the GAN optimization process can be combined with the opinions by multiple critics from different aspects, thus the zero-sum adversarial game rule proposed by Goodfellow et al. [7] has been changed to a multi-domain task. Furthermore, the Ad CycleGAN does not need to encode the labels into the training data therefore it simplifies the computation runtime for GAN optimizations.

The most significant merit of the Ad CycleGAN is that it improves the image translation accuracy to the target image category. In our literature review, most of the applications of GAN is to use this generative model for image data augmentation, but there is no guarantee of the generated images falling into the correct category. Therefore, the introduction of the pre-trained independent criterion becomes a unique impact of the Ad CycleGAN to the GAN architecture.

In addition, Ad CycleGAN can perform both image augmentation and image translation. Image augmentation means the input real images belongs to the same category as the expect synthetic outputs, e.g., from normal images to normal images with acceptable diversity, or from disease positive images to disease positive images with acceptable diversity. Currently, most of the GAN studies on medical images are focusing on image augmentation. The GAN models generate multiple synthetic samples including synthetic images for direct DNN optimization, image mask for improving image segmentation, and image feature maps for medical diagnosis and decision making. The applications of image translation are mainly for converting the images from one format to another format, such as from MR images to CT images, and from ultrasound images to CT images. However, all the available applications have not explored the task of converting images from the normal or healthy domain to a specific disease domain, which is crucial in medical research. Our experiment proves that the new Ad CycleGAN performs higher accurate heterogenous image translation (i.e., X→Y) than the original Cycle GAN model.

The Ad CycleGAN architecture provides the flexibility to add more external critics to control multiple aspects of GAN based image synthesis. As Khaldi Y et al. stressed the importance of image color control by GAN [30] and Creswell A et a. emphasized the proper image domain mapping for GAN image generation [31], the trade-off between model complexity and controllability is one of the main considerations for GAN based image generative models.

## 5. Conclusions

The findings in the experiments indicates that the newly proposed Ad CycleGAN can perform accurate medical image translation. We hope that this unique feature is helpful to solve the common class imbalance issues because the medical images containing rare or new disease information are both difficult to acquire and expensive for expert annotation. The successful applications of GAN for COVID-19 pattern detection and segmentation shows the feasibility to widely use DNN for computer assisted disease diagnosis and public health management. It also provides a new approach for rapid deployment of AI solutions for portable and wearable devices for COVID-19 or other public health challenges.

The future work on Ad CycleGAN will focus on two aspects. First, we will further improve the optimization objective function to ensure more control on the optimization process and minimize the side effects on the external criterion from synthesizing high-quality images like reducing the occurrence of artifacts on the synthetic images. More extra criterion can also be added to the objective to further control the characteristics of the generated images to the due domain. Second, we can develop a more sophisticated GAN architecture to extend the Ad CycleGAN design for more tasks. For example, the Ad CycleGAN can be optimized with the Pix2Pix architecture to precisely allocate the location of the synthetic patterns. Therefore, the application of the Ad CycleGAN can be extended to medical image segmentation. In conclusion, GAN provides a promising solution for the data greedy feature of deep neural networks. The new Ad CycleGAN provides more authentic images to augment the training of DNN with high performance and robustness. We believe this new technology will promote DNN-related technologies for medical diagnosis and decision-making, and it will ultimately help to enhance high-quality healthcare delivery.

## Figures and Tables

**Figure 1 sensors-22-09628-f001:**
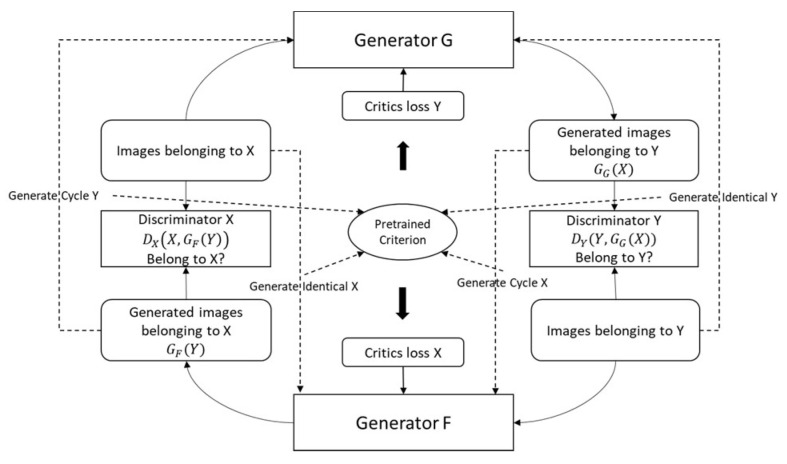
Ad CycleGAN Architecture.

**Figure 2 sensors-22-09628-f002:**
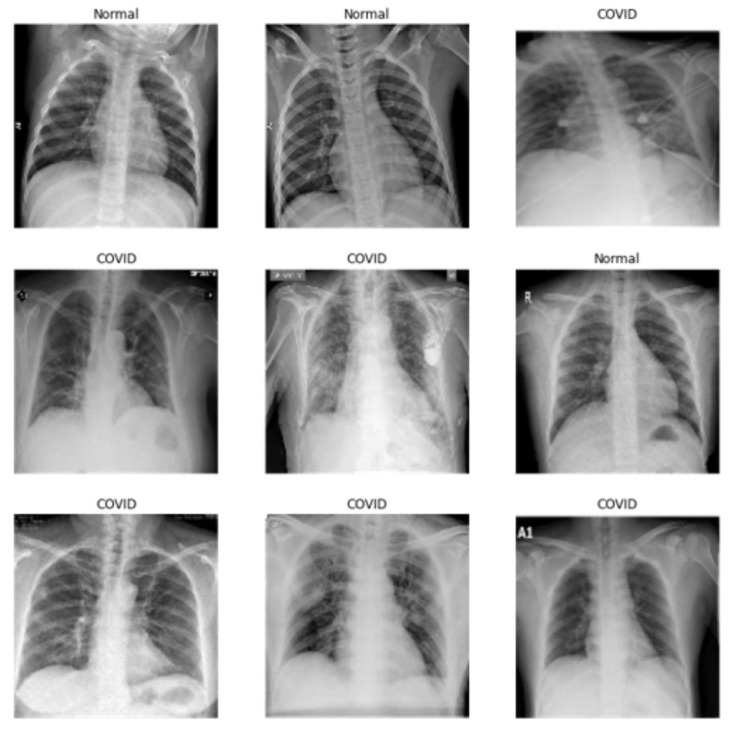
Original Chest X-ray Images.

**Figure 3 sensors-22-09628-f003:**
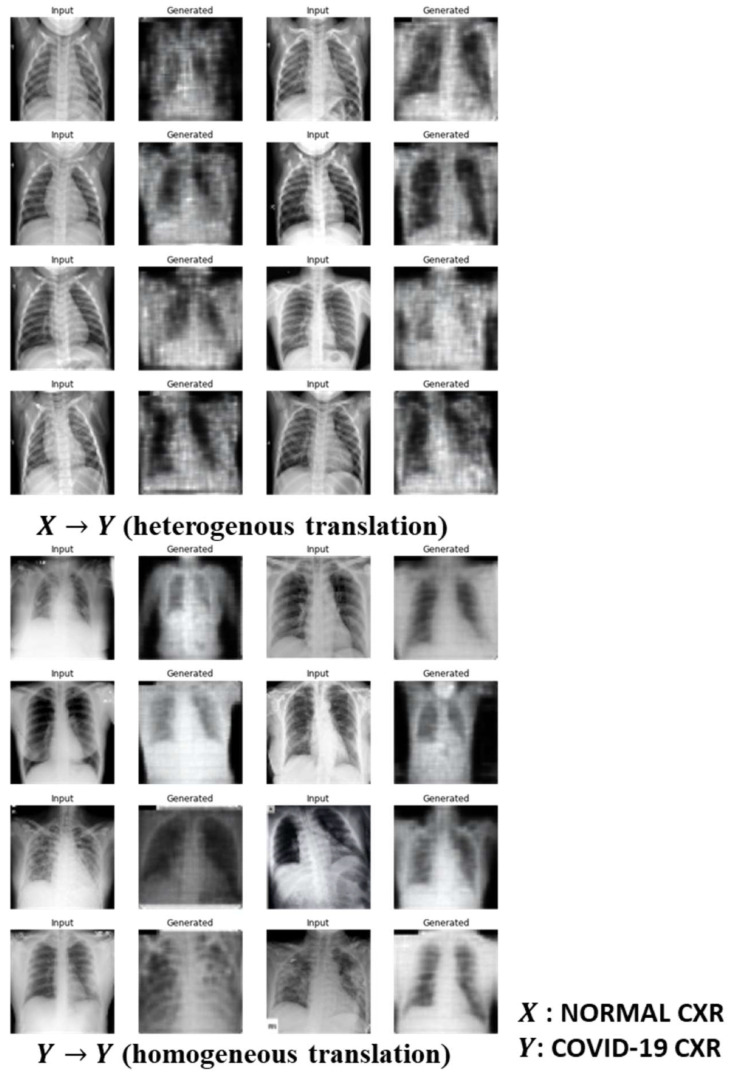
Image translation by Cycle GAN.

**Figure 4 sensors-22-09628-f004:**
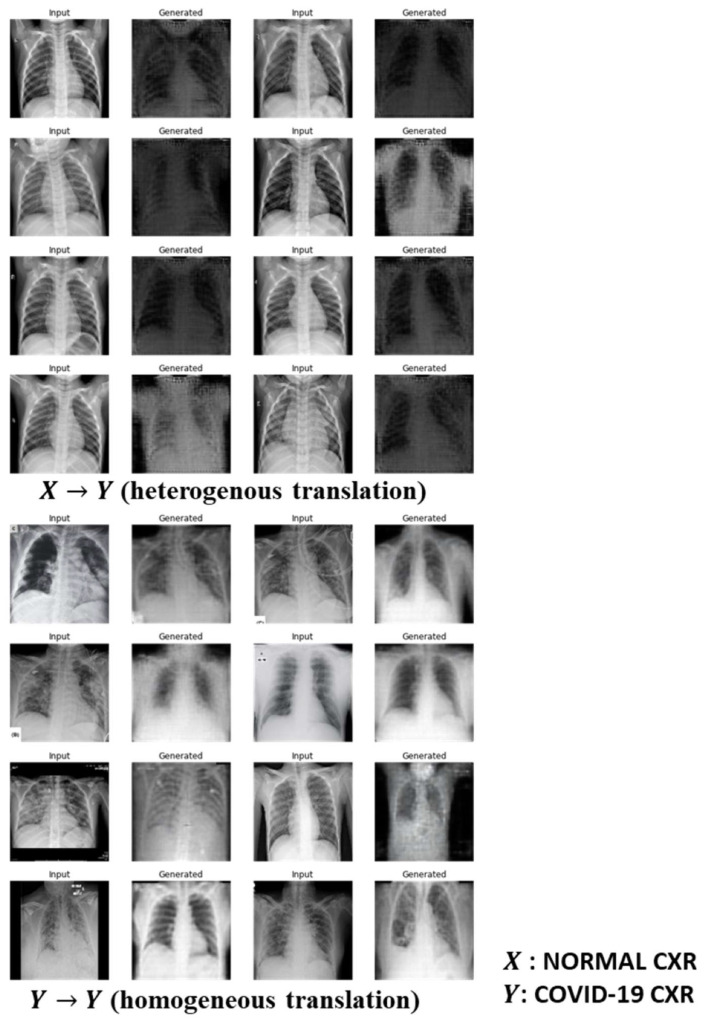
Image translation by Ad CycleGAN.

**Figure 5 sensors-22-09628-f005:**
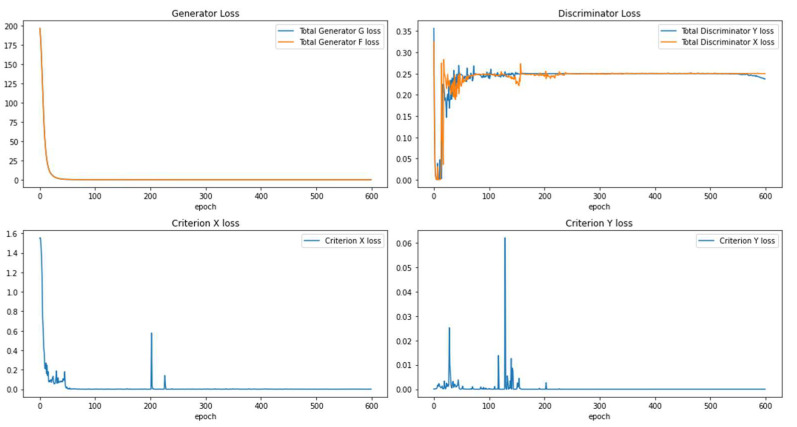
Optimization process of Ad CycleGAN.

**Table 1 sensors-22-09628-t001:** Quantitative metrics for the synthetic Images.

Model (Translation Direction)	MSE (std)	RMSE (std)	PSNR (std)	UIQI (std)	VIF (std)
Cycle GAN (X→Y)	3608.97 (1398.37)	58.77 (12.42)	12.97 (2.10)	0.81 (0.08)	0.12 (0.05)
Cycle GAN (Y→Y)	409.00 (495.043)	17.55 (10.035)	24.43 (4.398)	0.97 (0.029)	0.55 (0.050)
Ad CycleGAN (X→Y)	3750.71 (1789.51)	59.54 (14.30)	12.88 (2.12)	0.80 (0.13)	0.10 (0.03)
Ad CycleGAN (Y→Y)	435.84 (461.59)	18.73 (9.21)	23.59 (3.86)	0.97 (0.03)	0.52 (0.05)

**Table 2 sensors-22-09628-t002:** FID score and classification accuracy.

Model (Translation Direction)	FID (std)	**Accuracy**
Cycle GAN (X→Y)	$5.26\times 10^{-6}$ $\left(1.82\times 10^{-5}\right)$	0.9375
Cycle GAN (Y→Y)	$6.31\times 10^{-4}$ $\left(4.40\times 10^{-7}\right)$	1.0
Ad CycleGAN (X→Y)	$1.19\times 10^{-5}$ $\left(4.16\times 10^{-6}\right)$	0.9843
Ad CycleGAN (Y→Y)	$3.60\times 10^{-4}$ $\left(6.44\times 10^{-5}\right)$	1.0

## Data Availability

The COVID-19 CXR image dataset in this study is from an open source COVID-19 Radiography Database on Kaggle, which is accessible at: https://www.kaggle.com/datasets/tawsifurrahman/covid19-radiography-database (accessed on 29 August 2022).

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
