# Peer review of "Image Translation by Ad CycleGAN for COVID-19 X-Ray Images: A New Approach for Controllable GAN"

_sensors, 2022, doi:10.3390/s22249628_

Round 1

Reviewer 1 Report

In the presented work, the authors proposed a new generative model named adaptive cycle-consistent generative adversarial network, or Ad CycleGAN, to perform image translation between typical and COVID-19-positive chest X-ray images. Overall, the manuscript is well-written and organized. However, I have several comments in order to improve its quality further:

-         The main contributions should be added in the last part of the introduction.

- The paper’s organization should be added in the last part of the introduction.

-         Authors should add some perspectives in the last part of the conclusion.

-         Add a list of abbreviations used in the manuscript.

-         The merit of the proposed approach is supported by the results. But, I miss on the paper a bit more discussion on why these techniques were chosen for this problem and had not been considered before. This, however, is more of a nitpicking than a detrimental comment.

-         The authors must have an external and well-known validation dataset to evaluate the performance of the proposed approach.

-         The authors should compare their performance to previously published works on the same problem/data.

-         Some references related to GANs should be added to attract a broader readership. E.g., 10.1007/s12530-020-09346-1 and https://doi.org/10.48550/arXiv.1611.05644.

-         Uncertainties of the model should be reported.

-         What are the limitations of the present work?

Author Response

Author’s Response to Review Comments

I would like to thank the reviewer for the valuable review comments. I provide my feedback to the review comments one-to-one as below.

  1. The main contributions should be added in the last part of the introduction.

Response: The main contribution of this study is to introduce an external criterion to the current state-of-the-art GAN architecture (Cycle GAN) for image translation, which can ensure the generated synthetic images belong to both the correct image domain and the correct diagnosis class. This design will be easy to extend to other medical or non-medical data synthetization applications. 

The corresponding changes can be found in the last paragraph of Introduction.

  1. The paper’s organization should be added in the last part of the introduction.

Response: The paper’s organization is added as the second part of the last paragraph in the Introduction Section following the main contributions.

  1. Authors should add some perspectives in the last part of the conclusion.

Response: we add the following content in the last part of the conclusion.

The future work on Ad CycleGAN will focus on two aspects. First, we will further improve the optimization objective function to ensure more control on the optimization process and minimize the side effects on the external criterion from synthesizing high-quality images like reducing the occurrence of artifacts on the synthetic images. More extra criterion can also be added to the objective to further control the characteristics of the generated images to the due domain. Second, we can develop a more sophisticated GAN architecture to extend the Ad CycleGAN design for more tasks. For example, the Ad CycleGAN can be optimized with the Pix2Pix architecture to precisely allocate the location of the synthetic patterns. Therefore, the application of the Ad CycleGAN can be extended to medical image segmentation. In conclusion, GAN provides a promising solution for the data greedy feature of deep neural networks. The new Ad CycleGAN provides more authentic images to augment the training of DNN with high performance and robustness. We believe this new technology will promote DNN-related technologies for medical diagnosis and decision-making, and it will ultimately help to enhance high-quality healthcare delivery.

  1. Add a list of abbreviations used in the manuscript.

Response: the list of abbreviations is added under Section 5 Conclusions.

  1. The merit of the proposed approach is supported by the results. But, I miss on the paper a bit more discussion on why these techniques were chosen for this problem and had not been considered before. This, however, is more of a nitpicking than a detrimental comment.

Response: the proposed approach can ensure the generated synthetic images belong to both the correct image domain and the correct diagnosis class. It can be found in the last paragraph of Section 1 Introduction. It is also helpful to solve the common class imbalance issues because medical images containing rare or new disease information are both difficult to acquire and expensive for expert annotation. The corresponding content can be found in the first paragraph of Section 5 Conclusions..

  1. The authors must have an external and well-known validation dataset to evaluate the performance of the proposed approach.

Response: the dataset used in this study is the open-source the Kaggle COVID-19 Radiography dataset (https://www.kaggle.com/datasets/tawsifurrahman/covid19-radiography-database) that contains three classes: normal chest X-ray images, viral pneumonia chest X-ray images, and COVID-19 chest X-ray images.

The corresponding information can be found in the Data Availability Statement.

  1. The authors should compare their performance to previously published works on the same problem/data.

Response: In our literature review, the similar research on COVID-19 CXR image synthesis mainly uses accuracy as the main metric for GAN performance.

For example, Motamed S et al. used GAN as data augmentation to improve the COVID-19 CXR image classification accuracy from 0.81 to 0.84 (Ref 29). Morís DI et al. used CycleGAN to improve COVID-19 CXR image screening accuracy to about 0.90 (Ref 30). Both studies did not use GAN with effective controllable mechanism to guarantee the synthetic images falling into the COVID-19 positive category.

The corresponding content is added at line 348 – line 354.

  1. Some references related to GANs should be added to attract a broader readership. E.g., 10.1007/s12530-020-09346-1 and https://doi.org/10.48550/arXiv.1611.05644.

Response: The two papers mentioned in the comments have been included in the reference and quoted in the last paragraph of the Discussions Section.

Ref 31. Khaldi, Y.; Benzaoui, A.; A new framework for grayscale ear images recognition using generative adversarial networks under unconstrained conditions. Evolving Systems. 2021 Dec;12(4):923-34. DOI: 10.1007/s12530-020-09346-1.

Ref 32. Creswell, A.; Bharath, A.A.; Inverting the generator of a generative adversarial network. IEEE transactions on neural networks and learning systems. 2018, 30(7), 1967-74. DOI: https://doi.org/10.48550/arXiv.1611.05644.

Reviewer 2 Report

This paper tries to solve the dataset augmentation issue encountered in the medical imaging field by generating more realistic images using GAN. To overcome the shortcomings of the current GAN method, the authors proposed adding a pre-trained DNN as an independent criterion to evaluate the generated images. This idea is very interesting and useful. However, this manuscript can be improved from the following aspects:

1. Images. The images presented in figure 3 and figure 4 are too small to view. Could you authors show some zoomed-in views, especially for the pathology portions of the images? It would also be helpful if the authors could indicate the pathology portions in zoomed-in views in figure 2.

2. Pretrained neural networks (line 188 - 190). Could the authors review the details about the pre-trained neural networks (e.g., training methods, training datasets, evaluation methods, etc.)?

3. Training datasets (line 269 - 270). Could the authors describe the source and the details (e.g., acquisition conditions, patient conditions, etc.) of the training datasets used?

Author Response

I would like to thank the reviewer for the valuable review comments. I provide my feedback to the review comments one-to-one as below.

1 Images. The images presented in figure 3 and figure 4 are too small to view. Could you authors show some zoomed-in views, especially for the pathology portions of the images? It would also be helpful if the authors could indicate the pathology portions in zoomed-in views in figure 2.

Response: since our goal is to apply the Adaptive Cycle GAN (Ad CycleGAN) to perform image synthesis by translating the chest X-ray (CXR) images from the normal domain to the COVID-19 positive domain. We revise figure 3 (between line 291 and 292) and figure 4 (between line 293 and 294) to show the synthetic images by heterogeneous translation from normal CXR to COVID-19 positive CXR, and by homogeneous translation from COVID-19 positive CXR to COVID-19 positive CXR with feature augmentation. With this change, all individual images will be larger and clearer to view. 

Regarding the pathology portions, the pathological significant patterns can be in any position of the chest X-ray images. According to the report in The Lancet, the COVID-19 positive cases have special bilateral or unilateral multiple mottling and ground-glass opacity patterns on the CXR and CT images (Chen N et al., 2020, Ref 1). Unlike the image patterns, such as thoracic tumors having specific locations, the pathological patterns of COVID-19 pneumonia cannot be highlighted in CXR images. We revised Figure 2 from the 4X4 format to the 3X3 format. In addition, on the top of each CXR image, the COVID-19 positive image is labeled with “COVID” and the normal CXR image is labeled with “Normal”, we think this change can give the reader a clear comparison of the two classes of CXR images.

The corresponding changes can be found between line 291 and line 292 (Figure 3), between line 293 and line 294 (Figure 4), and between line 279 and line 280 (Figure 2)

  1. Pretrained neural networks (line 188 - 190). Could the authors review the details about the pre-trained neural networks (e.g., training methods, training datasets, evaluation methods, etc.)?

Response: the pretrained neural network used as the criterion is a residual network architecture with 3 residual modules. Each residual module has three convolutional layers with a skip connection from the first convolutional layers to the third one to ensure gradient flow when optimized by backpropagation. The residual modules are connected by batch normalization and max pooling layers to accordingly reduce the tensor size. It uses the sparse categorical cross entropy function as the loss objective, and it is optimized by the adaptive moment estimation (Adam) algorithm with the initial learning rate of   with 100 epochs. During the GAN optimization, the pretrained criterion estimates the input images with the output logits which can be combined to other loss terms. The training dataset is the Kaggle COVID-19 Radiography dataset (https://www.kaggle.com/datasets/tawsifurrahman/covid19-radiography-database) that contains three class: normal chest X-ray images, viral pneumonia chest X-ray images, and COVID-19 chest X-ray images. Note that the three-class DNN can also be used for the Ad CycleGAN optimization with the correct labels during GAN optimization.

The pretrained DNN is evaluated by its classification accuracy. Our pretrained model has an average accuracy of 98.5%.  The corresponding revised content is between line 204 to line 212.

  1. Training datasets (line 269 - 270). Could the authors describe the source and the details (e.g., acquisition conditions, patient conditions, etc.) of the training datasets used?

Response: the training dataset is the open-source the Kaggle COVID-19 Radiography dataset (https://www.kaggle.com/datasets/tawsifurrahman/covid19-radiography-database) that contains three classes: normal chest X-ray images, viral pneumonia chest X-ray images, and COVID-19 chest X-ray images. The training of Ad CycleGAN follows the training procedure of Cycle GAN. The corresponding changes can be found between line 277 to line 278.

Round 2

Reviewer 1 Report

The responses are very convincing. The paper can be accepted in its current form.